# mTOR Inhibition Is Effective against Growth, Survival and Migration, but Not against Microglia Activation in Preclinical Glioma Models

**DOI:** 10.3390/ijms24129834

**Published:** 2023-06-07

**Authors:** Lucia Lisi, Michela Pizzoferrato, Gabriella Maria Pia Ciotti, Maria Martire, Pierluigi Navarra

**Affiliations:** Department of Healthcare Surveillance and Bioethics, Section of Pharmacology, Catholic University Medical School, Fondazione Policlinico Universitario A. Gemelli-IRCCS, 00168 Rome, Italy; michela.pizzoferrato@gmail.com (M.P.); gabriellamariapia.ciotti@unicatt.it (G.M.P.C.); maria.martire@unicatt.it (M.M.); pierluigi.navarra@unicatt.it (P.N.)

**Keywords:** mTOR, rapamycin, sapanisertib, glioblastoma, microglia

## Abstract

Initially introduced in therapy as immunosuppressants, the selective inhibitors of mTORC1 have been approved for the treatment of solid tumors. Novel non-selective inhibitors of mTOR are currently under preclinical and clinical developments in oncology, attempting to overcome some limitations associated with selective inhibitors, such as the development of tumor resistance. Looking at the possible clinical exploitation in the treatment of glioblastoma multiforme, in this study we used the human glioblastoma cell lines U87MG, T98G and microglia (CHME-5) to compare the effects of a non-selective mTOR inhibitor, sapanisertib, with those of rapamycin in a large array of experimental paradigms, including (i) the expression of factors involved in the mTOR signaling cascade, (ii) cell viability and mortality, (iii) cell migration and autophagy, and (iv) the profile of activation in tumor-associated microglia. We could distinguish between effects of the two compounds that were overlapping or similar, although with differences in potency and or/time-course, and effects that were diverging or even opposite. Among the latter, especially relevant is the difference in the profile of microglia activation, with rapamycin being an overall inhibitor of microglia activation, whereas sapanisertib was found to induce an M2-profile, which is usually associated with poor clinical outcomes.

## 1. Introduction

The mammalian target of rapamycin (mTOR), a 289 kDa atypical serine/threonine protein kinase, is a member of the phosphatidylinositol 3-kinase (PI3K)-related kinase superfamily. mTOR is the catalytic subunit of two different complexes, mTOR complex 1 (mTORC1) and mTOR complex 2 (mTORC2) [1]. All mTOR inhibitors currently available for clinical use are selective blockers of mTORC1.

Overall, evidence shows that mTOR plays a central role in the synthesis of key cellular proteins, which are important in many aspects of cell growth and proliferation [2]. Dysregulation of mTOR signaling occurs in a variety of human cancers. Tumors harboring up-regulation of the mTOR pathway show high susceptibility to the pharmacological inhibition of mTOR in both in vitro and in vivo models [3]. Because of mTOR involvement in the control of tumor progression, the mTOR inhibitors, aside from their well-established use as immune suppressants, have also been developed and approved in oncology indications. The selective mTOR inhibitor, everolimus exerts its action by binding to the intracellular receptor FKBP12, which is followed by the inhibition of cell growth and subsequent cytotoxic effects [2]. This finding is consistently observed in preclinical models [4]. Everolimus is approved for the treatment of solid tumors, including renal cell carcinoma (RCC), hormone receptor-positive advanced breast cancer and neuroendocrine pancreatic tumors [5]. Moreover, everolimus is currently under scrutiny in preclinical investigations and in the early phase of clinical development for the treatment of non-small cell lung cancer (NSCLC). Meanwhile, everolimus is given for compassionate use in late-stage NSCLC, in patients with no further therapeutic options [5].

mTORC1 inhibitors showed antitumor activity with in vitro and in vivo glioma models [6]. In addition to direct antitumor activity, there is further evidence supporting the efficacy of the mTOR blockade in GBM: in isocitrate dehydrogenase (IDH)-wildtype glioblastoma (GBM), PI3K/mTOR is one of the frequently altered molecular pathways, due to the loss of the tumor suppressor phosphatase and tensin homolog [7,8]. In addition, the inhibition of mTOR polarizes glioma-associated macrophages/microglia towards a proinflammatory phenotype, preventing the pro-tumor phenotype with in vitro models [9,10]. Despite such promising preclinical data, mTORC1 inhibitors failed to show efficacy in clinical trials in GBM patients, even in combination with EGFR inhibitors [11]. Such lack of efficacy might be related to a feedback loop that hyper-activates Akt and mTORC2 downstream of the mTOR blockade. Other possible resistance mechanisms may play a role involving infiltrating microglial cells. In fact, it should be considered that mTOR inhibitors act both on tumor and on immune cells; thus, one can hypothesize that the putative anti-tumor efficacy of mTOR inhibitors on cancer cells might be counterbalanced by their suppressive effects on immune cells, thereby building an immunosuppressive environment that facilitates tumor progression [12,13].

To overcome the above-described drawbacks, new molecules blocking both mTORC1 and mTORC2 complexes are currently being developed [14]. Sapanisertib (SAP, also referred to as MLN0128, INK128, TAK-228) is a catalytic mTORC1/2 kinase inhibitor developed with the aim of suppressing the activation of Akt by mTORC2, which follows the selective inhibition of mTORC1 [15]. In the literature, there are few data regarding the BBB permeability of SAP. Parkhurst and collaborators show that in mice SAP is rapidly absorbed and crosses the blood-brain barrier in concentrations exceeding the therapeutic threshold required to inhibit the mTOR pathway in vitro [16]. Using SAP on in vitro models of human GBM cell lines, Heinzen and collaborators found conflicting results, with the favorable inhibitory effect on cancer cell growth being counterbalanced by the protective action against hypoxia-related cell death [17]. A combination of mTORC1/2 and MEK inhibitors, tested on a series of genetically characterized glioblastoma cell lines, showed that such combination even stimulates glioma cell growth [18]. Taken together, these pre-clinical findings allow us to foresee that the inclusion of non-selective mTOR inhibitors in treatment strategies for GBM might be a challenging task. Moreover, preclinical models that adequately mirror the conditions of the tumor microenvironment (looking at the contribution of microglia infiltrating cells) remain unexplored so far [17]. 

Concerning the clinical setting, SAP showed a suitable safety profile and preliminary antitumor activity, either alone or in combination with paclitaxel or trastuzumab, in several advanced solid tumor types [19]. SAP is currently under investigation in a large array of tumors including—among others—hepatocellular carcinoma (HCC), thyroid tumors, gliosarcoma and glioblastoma. To date, 44 clinical trials are listed in clinicaltrials.gov (accessed on 2 June 2023) which investigates SAP in oncology indications; in particular, two studies are underway in GBM patients. A phase I trial investigates the safety and the best dose of SAP given in combination with bevacizumab in patients with glioblastoma, or other solid tumors that have spread and not responded to standard treatments [20]. Another pilot phase I trial investigates SAP penetration into brain parenchyma, and its preliminary efficacy in patients with GBM undergoing surgery [21]. The results of these studies are not available yet.

Within this framework, in the present study, we carried out an extensive series of experiments comparing SAP with the selective mTOR inhibitor rapamycin in two different glioma cell lines as well as in models of GBM–microglia interaction.

## 2. Results

### 2.1. Effects of mTOR Inhibitors on U87MG Cells

#### 2.1.1. Effects of Rapa and SAP on Factors Acting Downstream and Upstream mTOR Complexes

In experiments that look at the total and phosphorylated mTOR, after 2 h of exposure to 10 nM Rapa and 10 nM SAP, the levels of mTOR phosphorylation at Ser 2448 were significantly reduced and 10 nM SAP was also able to reduce the total mTOR content (Figure 1A). Looking at the effects on the above-described factor, 4EBP1, i.e., a well-established factor downstream the mTOR signaling cascade [22], 4EBP phosphorylation is reduced with 10 nM Rapa and 10 nM SAP. Whereas 10 nM Rapa and 10 nM SAP were able to reduce the levels of phospho-p70S6K Thr 389 and no significant effect was reported on the phospho-p70S6K at Ser 411 (Figure 1A,B). Looking at factors upstream for both mTORC1 and -2 complexes, Rapa and SAP had no significant effects on the TSC2 levels (Figure 1B). Similarly, the levels of total AKT, a factor downstream of mTORC2 and upstream of mTORC1 [22], were not modified by both Rapa and SAP (Figure 1B), but 10 nM Rapa was able to increase the phosphorylation at Ser 473 of the AKT, and 10 nM SAP was able to reduce the phosphorylation at Ser 473 of the AKT (Figure 1B). 

It is not surprising that the phosphorylation of AKT at Ser 473 increased after 2 h of Rapa treatment. This is very likely due to the release of the negative feedback loop formed by the phosphorylation and inhibition of IRS1, and thereby PI3K, by S6K [23].

The normalization data for pmTOR/total mTOR and pAKT/total AKT were reported in Appendix A. 

#### 2.1.2. Effects of Rapa and SAP on Viability of U87MG Cell Line

A second set of experiments was carried out to evaluate cell viability after a challenge with Rapa or SAP. After the 48 h of treatment, we observed a dose-dependent reduction in cell viability, as measured by the XTT assay. In particular, 100 nM Rapa reduced, by about 10%, U87MG viability, whereas 10 and 100 nM SAP reduced U87MG viability by 10 and 50%, respectively (Figure 2A). Such reduction in cell viability was maintained after 72 h (Figure 2B), and up to 6 days (Figure 2C) of treatment. Likewise, protein content was reduced with both Rapa and SAP after 72 h and 6 days of exposure. In particular, after 72 h, the level of protein was reduced by about 10% after exposure to 1 nM Rapa, and by about 30% after 10 nM and 100 nM Rapa (Figure 2D). Similar to Rapa, SAP 10 and 100 nM reduced protein levels by 10 and 50%, respectively (Figure 2D). After 6 days of treatment, the protein content was reduced by more than 60 and 80% after exposure to 100 nM Rapa and SAP, respectively (Figure 2E). Finally, after 72 h of exposure, both Rapa and SAP were able to increase LDH release, expressed as a percent of extracellular LDH on total LDH. These effects were observed at all concentrations tested, with no dose-dependent relationship (Figure 2F).

Finally, experiments were conducted where Rapa and SAP were co-administered for 6 days. The results are described in Appendix A.

#### 2.1.3. Effects of Rapa and SAP on Migration of U87MG Cell Line

As far as cell migration was concerned, cells were exposed for 16 h to a fixed dose of Rapa or SAP (10 nM). Apart from the vehicle (1% FBS in medium), medium with 10% FBS was also included in the experiments as a positive control, since this treatment increased per se the number of migrating cells. We found that both mTOR inhibitors were able to significantly reduce the number of migrating cells compared to the vehicle (Figure 3). 

To elucidate the mechanism of action underlying the reduction in migration induced by mTOR inhibitors, we investigated the gene expression of ADAM 10 and ADAM 17, two factors promoting tumor invasiveness by degradation of the extracellular matrix and by inducing the epithelial–mesenchimal transition (EMT) [24,25]. After 24 h of treatment, Rapa (10 and 100 nM) and SAP (100 nM but not 10 nM) were able to significantly increase the gene expression of both ADAM 10 and 17 (Figure 4A,B). After 48 h of exposure, the level of ADAM 17 was significantly reduced by 10 nM Rapa and SAP (Figure 4E), whereas the levels of ADAM 10 remained elevated (Figure 4D). 

At the same time points (24 and 48 h), TGFβ gene expression was studied. After 24 h of treatment, only SAP (100 nM but not 10 nM) was able to significantly increase the gene expression of TGFβ (Figure 4C). This effect persists even after 48 h of treatment (Figure 4F).

In the experiments looking at proteins involved with EMT, after 2 h of exposure the levels of pS6 ribosomal protein were significantly reduced with 10 nM of both RAPA and SAP (Figure 5A,B) and, in parallel, the level of total S6 was increased (Figure 5A,B). In addition, other proteins involved in EMT were studied (N-cadherin, pEGF receptor and IKBα) but no significant modifications were reported (Figure 5A).

#### 2.1.4. Effects of Rapa and SAP on Autophagy and Apoptosis Proteins

mTOR is a key player in autophagy [26]. In this study, two of the main proteins involved in the autophagy mechanism were studied, namely LC3A and beclin [27]. As expected, 10 nM Rapa modified the ratio of LC3AI/LC3AII by increasing LC3AII levels. The same effect was observed with 10 nM SAP (Figure 6A,B). At variance, the two drugs had opposing effects on the beclin protein; in fact, beclin levels were significantly reduced and increased by Rapa and SAP, respectively (Figure 6A,C).

To investigate whether mTOR inhibitors activate mechanisms of apoptosis, we investigated the cleaved caspase 3 (Figure 7A,B). In particular, we studied the ratio between the P17 and P19 forms. In line with an induction of apoptosis, the P17 form increases in the presence of Rapa at 10 nM and SAP at 1 and 10 nM. In addition, Rapa at 10 nM tends to reduce, while SAP at 10 nM significantly reduces p21 levels (Figure 7C,D), a protein closely related to caspase 3 activity and therefore apoptosis [28,29].

The main experiments carried out on U87MG cells were repeated in another human glioma cell line, T98G, obtaining similar results. Data are explained in the Appendix A.

### 2.2. Effects of mTOR Inhibitors in Microglia-Glioma Interaction Model

A preliminary series of experiments was carried out on human microglia CHME-5 cells [30] to test the putative toxicity of both Rapa and SAP under basal conditions. Real time dose–range experiments were carried out. Both viable and non-viable cells were measured. Under baseline conditions, the number of viable cells decreased, and conversely the number of non-viable cells increased, after 24 h of exposure. After 42 h of exposure, the number of viable cells reached a plateau, while the number of non-viable cells was further increased (Figure 8). Both Rapa (1 and 10 nM) and SAP (1–100 nM) had no effect whatsoever on this paradigm (Figure 8).

Our group has previously investigated the L-arginine (L-ARG) metabolic pathways in microglial cells, taken as a marker of microglia polarization [10,31,32,33,34]. Here we tested the effects of Rapa and SAP on nitrite and urea production, considered as end-products of the L-ARG metabolism and indicators of M1 or M2 microglia polarization, respectively. In the presence of an inflammatory stimulus, i.e., the mixture of cytokines TII, Rapa (1 and 10 nM) was able to reduce nitrite production while at the same time reducing urea release, at a 10 nM concentration (Figure 9A,B). Looking at the effects of SAP, 1 and 10 nM SAP did not modify nitrite production elicited by TII, while nitrites were significantly reduced at the higher concentration tested (Figure 9A,B). Conversely, 100 nM SAP further increased urea production elicited by TII (Figure 9A,B). 

In a validated model of the microglia–glioma interaction, we tested the effects of both Rapa and SAP. Two different settings were used: (1) in order to mimic the early stage of the GBM pathology, we used a basal conditioned medium (B-CM) harvested from U87MG; and (2) to mimic a later stage of pathology, we used a pre-stimulated conditioned (PS-CM) medium that was harvested from U87MG [9]. Rapa was able to significantly reduce urea release elicited by B-CM, whereas 1 and 10 nM of SAP did not modify the levels of urea production. On the opposite, 100 nM SAP significantly increased (by about 50%) urea production elicited by B-CM (Figure 10A). In the presence of PS-CM, Rapa did not modify urea release, but again 100 nM SAP significantly increased the levels of urea elicited by PS-CM (Figure 10B). In order to confirm the pro-M2 effect of SAP, urea levels were measured in CHME-5/U87MG co-culture experiments. SAP, at 100 nM, significantly increased urea levels after 48 h of treatment (Figure 10C).

## 3. Discussion

In the present study, we carried out an extensive series of experiments comparing SAP and Rapa in two different glioma cell lines, as well as in models of GBM–microglia interaction.

Overall, we could distinguish between experimental paradigms where the two agents had similar effects and those where the effects were diverging or opposing. Within the first subset of results, we could also appreciate differences in potency, although, in the context of effects consistent among each other. The following discussion will deal mostly with those paradigms where we observed diverging or opposing effects between Rapa and SAP. 

Both inhibitors were able to reduce cell viability. However, we found differences in potency and in the onset of action. In fact, while the higher concentrations of both drugs are almost equally effective, Rapa at 10 nM—but not SAP at 10 nM—shows toxicity from 72 h onwards, both on protein content and on the levels of released LDH. After 6 days of exposure, 10 nM SAP also achieved the same effects on protein content and LDH release. These differences might be related to different subsets of kinase activities modulated downstream for mTOR inhibition. In fact, various authors showed that the more kinases that are inhibited in cancer cells, the more prolonged exposures and higher concentrations of inhibitors are required [35,36]. 

Concerning autophagy, we found that both mTOR inhibitors are effective in increasing the microtubule-associated protein 1A/1B-light chain 3 (LC3) II level. The observed increase in LC3-II induced by mTOR inhibitors, which accompanies the accumulation of autopha gosomes, correlates well with the notion that the blockage of mTOR induces autophagy [37]. Looking at another important protein involved in autophagy, the mammalian orthologue of yeast Atg6 Beclin 1, we found that Rapa reduces and—on the opposite—SAP increases beclin 1 levels. Thus, the diverging effects of Rapa and SAP observed in glioma cells after 6 days of treatment can be well explained by the fact that SAP, by increasing both LC3A II and beclin1, induces a full activation of autophagy, whereas the reduction in beclin-1 production, associated with Rapa treatment, tends to counter-balance the increase in LC3A II, resulting in a lower level of autophagy. 

To promote tumor proliferation, tumor cells interact with nearby non-cancerous cells, including endothelial, epithelial, immune and glioma stem cells (GSCs), which altogether constitute the so-called tumor microenvironment [38]. Astrocytes and macrophages/microglia, defined as tumor-associated macrophages (TAMs), significantly contribute to enhance the tumor proliferation, as well as its capability to infiltrate nearby tissues. In the case of GBM, microglia represent more than 30% of the entire mass [39]. Different phenotypes may be present within the tumor [13], but microglia cells express mostly pro-tumor markers, defined as M2-profile. To investigate the profile of microglia activation, our group focuses on the analysis of the L-arginine (L-ARG) metabolic pathways [10,30,31,32,33]. In fact, L-ARG is a substrate for two different enzymes, arginase (ARG) and oxide nitric synthase (iNOS): L-ARG catabolism, through iNOS, produces nitric oxide (NO) and L-citrulline, whereas its catabolism via ARG yields urea and ornithine. Since the two enzymatic pathways share the same substrate, an excessive iNOS activity can possibly reduce the amounts of L-ARG, which yield urea through the ARG pathway and vice versa [40]. In this paradigm, mTOR inhibitors show different effects: Rapa reduces both NO and urea elicited by TII, whereas SAP reduces NO but increases the urea level. In addition, the increase in urea is stimulus-independent, since SAP is able to potentiate the increase in urea elicited by TII, B-CM and PS-CM. This finding suggests that a high concentration of SAP might polarize microglia cells toward an M2 pro-tumor phenotype. Further research is needed to clarify this point, in light of potential drawbacks in the clinical setting. 

At variance with previously discussed paradigms, where the effects of Rapa and SAP were diverging or different in potency/timing, the two drugs show similar effects on migration. However, this point deserves to be discussed because of the unexpected results about the potential mechanism of action of migration antagonism. In fact, we hypothesized that the reduction in migration might involve metalloproteinase (MMPs) activity, since the extracellular matrix (ECM) remodeling is mainly due to the activity of MMPs, a family of zinc-dependent endopeptidases that degrade the extracellular matrix components and cell surface proteins, affecting proliferation and adhesion mechanisms [41]. Among MMPs involved in glioblastoma progression, the ADAMs family stands out; it has been consistently shown that glioblastoma overexpresses mainly two members of the ADAMs family, ADAM 10 and ADAM 17 [24,25], suggesting a strong association between these two glycoproteins and glioblastoma progression [42,43] Based on the above evidence, we hypothesized to observe a decrease in ADAM 10 and ADAM 17 gene expression after exposure to the mTOR inhibitors. Instead, we found a significant increase in both metalloproteinase gene expressions. Therefore, we concluded that other factors contribute to the reduction of migration promoted by mTOR inhibitors. We then investigated several factors (EGF receptor [44], NFkB pathway [45], N cadherin proteins [46]) and we found a significant reduction in phosphorylated S6 protein. S6 is associated with the migration of glioma cells [47,48].

A final analysis can be performed by comparing the effects of drugs on different lines of glioblastoma (U87MG vs. T98G). In general, on both lines an antiproliferative effect (measurable as a reduction both in metabolic activity and in quantity of proteins) and a migration reduction effect were reported (Figure 2 and Figure 3 vs. Appendix A).

## 4. Materials and Methods

### 4.1. Cell Cultures 

Human U87MG and T98G glioma cell lines were kindly provided by professor Grazia Graziani (Tor Vergata University, Italy Rome). The human microglia cell line (CHME-5; RRID:CVCL_5J53) was kindly provided by professor Pierre Talbot. U87MG, T98G and CHME-5 were cultured in DMEM High Glucose (Corning, New York, NY, USA) supplemented with 10% FBS (Gibco, Thermo Fisher Scientific Inc., Waltham, MA, USA) and 100 U/mL penicillin–streptomycin at 37 °C in a 5% CO_2_ environment. Cells were splitted once a week when 90–100% confluence was reached. 

### 4.2. Conditioned Media

Conditioned media from U87MG was prepared both in basal and prestimulated conditions. In particular, basal conditioned medium (B-CM) was prepared treating U87MG for 4 h with medium containing 1%FBS. Then, 3 washes with phosphate buffered saline (PBS) were performed and fresh 1%FBS medium was added for an additional 24 h. After this time, the medium was collected, centrifuged for 5 min at 1100 rpm and then frozen at −80 °C. Pre-stimulated conditioned medium (PS-CM) was prepared treating U87MG for 4 h with medium containing 1%FBS and a mixture of cytokines (10 ng/mL TNFα, 10 ng/ ml IL1β, 10 UI/mL hIFNγ (TII)). Then, 3 washes with phosphate buffered saline (PBS) were performed and fresh 1%FBS medium was added for an additional 24 h. After this time, the medium was collected, centrifuged for 5 min at 1100 rpm and then frozen at −80 °C. 

### 4.3. Chemical Reagents

Rapamycin (RAPA) was purchased from Sigma-Aldrich (St. Louis, MO, USA) and dissolved in DMSO to obtain a stock solution of 10 mM Sapanisertib (SAP) was purchased from Merck & Co (Kenilworth, NJ, USA) and dissolved in DMSO to obtain a stock solution of 10 mM. TNFα, IL1β and IFNγ were purchased from R&D system (Minneapolis, MN, USA). In 6 day experiments, both drugs are replaced fresh after 72 h of treatment.

### 4.4. Nitrite Assay

After 24 h from splitting, CHME-5 was plated at a density of 20,000 cells/well and treated for 72 h with TII, in medium containing 1%FBS, and different concentrations of Rapamycin (1–10 nM) or Sapanisertib (from 1 nM to 100 nM). After 72 h of incubation, medium was collected, and nitrite amount was measured in 80 μL of medium after adding 40 μL Griess Reagent (Sigma-Aldrich, St. Louis, MO, USA). The absorbance was measured at 550 nm in a spectrophotometric microplate reader (PerkinElmer Inc., Waltham, MA, USA). A calibration curve (0–100 μM) was generated using NaNO_2_ (Sigma-Aldrich, St. Louis, MO, USA) as standard and all the data were normalized by protein quantitation (measured with Bradford’s method, using bovine serum albumin as standard).

### 4.5. Urea Assay

Cells were plated at a density of 20,000 cells/well and urea levels were measured after 48 h of treatment. Urea assay was performed also with CHME-5 treated with TII for 72 h. In all these cases, urea levels were detected by the QuantiChrom Urea Assay kit (BioAssay System, Hayward, CA, USA) according to the manufacturer’s instructions. Two hundred μL Urea Reagent (BioAssay system, Hayward, CA, USA) were added to 50 μL culture medium, and the absorbance measured at 430 nm in a spectrophotometric microplate reader (PerkinElmer Inc., MA, USA) after 50 min of incubation at room temperature under slow agitation conditions. A standard curve was generated in the range of concentrations 0–100 μg/mL using Urea as standard, and all the data were normalized proteins.

### 4.6. Cell Death and Cell Viability Assays

Cell death and cell viability were measured in real time with CHME-5, plated at a density of 20,000 cells/well, in three different conditions: (a) basal conditions; (b) in presence of TII); (c) in presence of conditioned medium (B-CM or PS_CM as indicated in figure legends). The drug concentrations selected were 1 nM and 10 nM for Rapa and 1 nM, 10 nM and 100 nM for SAP. CellTox™ Green Cytotoxicity Assay kit (Promega, Madison, WI, USA) and RealTime-Glo™ MT Cell Viability Assay kit were used, according to the manufacturer’s instructions. After 48 h of treatment, cell death and cell viability were measured as a function of luminescence and fluorescence at 485–535 nm.

### 4.7. Viability Assays

XTT (TACS^®^ XTT Cell Proliferation Assay Kit, Trevigen^®^-, Gaithersburg, MD, USA) was performed, according to the manufacturer’s procedure. Briefly, U87MG and T98G were plated in a 96-well plate at a density of 5000 and 10,000 cells/well, depending on the treatment duration. After 24 h from plating, culture medium was removed and cells were treated with different concentrations of Sapanisertib (ranged from 10 pM to 100 nM) and Rapamycin (ranged from 1 nM to 100 nM), both dissolved in medium containing 1% FBS. XTT assay was performed at 48, 72, 144 (6 days) h. Cell viability was assessed by measuring absorbance at 490 nm using a microplate photometer (Victor 4, PerkinElmer, Waltham, MA, USA) and was expressed as the percentage of cell viability in relation to the untreated controls. 

### 4.8. Bradford Assay

Bradford protein assay (Quick Start™ Bradford Protein Assay Kit, Biorad, Hercules, CA, USA) was performed to quantize protein levels. Cells were treated as explained above and lysed in 100 μL Triton. In total, 10 μL of each treatment was used to calculate the amount of protein per well. BSA was chosen as standard to develop a calibration curve ranged from 1 mg/mL to 0 mg/mL and the protein amount was measured as a function of absorbance at 570 nm using a microplate photometer (Victor 4, PerkinElmer, Waltham, MA, USA). Results were expressed as a total μg of proteins per well. 

### 4.9. Toxicity Assays

LDH assay (CytoTox 96^®^ Non-Radioactive Cytotoxicity Assay, Promega, Madison, WI, USA) was performed, following manufacturer’s instructions. U87MG and T98G were plated and treated with the same paradigm used for XTT assay and extracellular and intracellular LDH amount was read by measuring absorbance at 490 nm with a microplate photometer (Victor 4, PerkinElmer, Waltham, MA, USA) at 72 and 144 h. Extracellular LDH was measured in the medium released from cells, while intracellular one was measured from cell lysate. Results refer to the percentage of extracellular LDH on total LDH (where total is calculated as extracellular + intracellular content). 

### 4.10. Co-Culture Assay

To confirm more consistently the microglia–glioma interaction, a co-culture assay was performed. Both CHME-5 and U87MG were plated at a density of 100,000 cells/well in a 12 well transwell plate (Corning, New York, NY, USA). CHME-5 was cultured for 24 h in 10% FBS medium, while U87 in 1% FBS medium. After 24 h culture medium was replaced with medium 1% FBS containing SAP 1 and 10 nM, both for CHME-5 and U87 MG. Urea levels were measured after 48 h treatment, using QuantiChrom™ Urea Assay Kit (BioAssay System, Hayward, CA, USA).

### 4.11. Cell Migration Assay

Cells were plated in a filter at a density of 20,000 cells/well in a 12 well transwell plate (Corning, New York, NY, USA) in starvation conditions (serum-free medium). In the bottom of the well, treatments (SAP or Rapa) were added in 10%FBS medium at a final concentration of 10 nM. After 18 h at 37 °C, cells were rinsed with PBS with Ca^2+^ and Mg^2+^ and fixed with 4% formaldehyde. After washing in PBS with Ca^2+^ and Mg^2+^ and Methanol 100×, cells were colored with Giemsa, previously diluted 20 folds in distilled water, for 45 min. Then, cells were washed with distilled and acidulated water and the well membrane was fixed on a slide. Later, cells were counted using a microscope (Zeiss Axiofot, East Lyme, CT, USA).

### 4.12. RNA Extraction and Quantification

Both U87MG and T98G were plated in a 6 well plate at a density of 500,000 cells/well. After 24 h, cells were treated with Rapa (10 nM and 100 nM) and SAP (10 nM and 100 nM) for both 24 and 48 h and 6 days. At these time points, cells were lysed in Trizol solution (TRI Reagent^®^, Sigma-Aldrich, St. Louis, MO, USA) and RNA extraction was fulfilled. After this procedure, the total amount of the extracted RNA was measured with Qubit Fluorometer (Invitrogen, Carlsbad, CA, USA), according to the manufacturer’s instructions, and 500 ng of RNA were retro-transcripted for each sample using the PrimeScript™ RT Reagent Kit, following the manufacturer’s procedure. After retro-transcription, DNase/RNase-free distilled water (UltraPure™ DNase/RNase-Free Distilled Water, Invitrogen, Carlsbad, CA, USA) was added to each sample to obtain a final concentration of 10 μg/mL of cDNA.

### 4.13. RT-PCR

Quantitative changes in mRNA levels were estimated by real-time PCR (Q-PCR) using the following cycling conditions: 35 cycles of denaturation at 95 °C for 20 s; annealing and extension at 60 °C for 20 s; using the Brilliant III Ultra-Fast SYBRGreen QPCR Master Mix (Agilent Technologies, Santa Clara, CA, USA). PCR reactions were carried out in a 20 μL reaction volume in AriaMx Real-time PCR (Agilent Technologies, Santa Clara, CA, USA). The primers used for evaluation of gene expression are: β actin C12 F (5′-ACG TTG CTA TCC AGG CTG TGC TAT-3′) and D01 R (5′-TTA ATG TCA CGC ACG ATT TCC CGC-3′); h ADAM17 R1744 (5′-AAG GAC TGT TCC TGT CAC TGT-3′) and F1612 (5′-GTT TGT GGG AAC TGC AGG GT-3′); hADAM 10 R507 (5′-ATA CTG ACC TCC CAT CCC CG-3′) and F 375 (5′-TTC TCC CTC CGG ATC GAT GT-3′). hTGFβ C08 F (5′-CAG TCA CCA TAG CAA CAC TC-3′) and C09 R (5′-CCT GGC CTG AAC TAC TAT CT-3′)

### 4.14. Western Immunoblot

Both U87MG and T98G were plated at a density of 15,000 cells/ cm^2^ and treated up to 6 days, refreshing culture medium on the third day. At day 6, cells were scraped in PBS without Ca^2+^ and Mg^2+^ and centrifuged at 1100 rpm for 5 min. Then, cells were lysed in RIPA buffer [1 mM EDTA (Cat. No.: E7889), 150 mM NaCl (Cat. No.: S9888, 1% igepal (Cat. No.: I3021), 0.5% sodium deoxycholate (Cat. No.: D-6750), 50 mM Tris–HCl, pH 8.0 (Cat.No.: T-3038) (Sigma-Aldrich, St. Louis, MO, USA), and 0.1% sodium dodecyl sulphate, SDS, (Cat. No.:1610416—Bio-Rad, Hercules, CA, USA)] containing protease inhibitor cocktail diluted 1:250 (Cat. No.: P8340—Sigma–Aldrich, St. Louis, MO, USA). The total protein amount was measured following the Bradford protein assay, as described above. Seven % polypolyacrylamide home-made and precast gels (Invitrogen, Carlsbad, CA, USA) were used for our experiments. Briefly, 80μg of proteins/sample were mixed with 4× Bolt™ LDS Sample Buffer (Cat. No.: B0007—Novex, Carlsbad, CA, USA) and 10× Bolt™ Sample Reducing Agent (Cat. No.: B0009—Novex, Carlsbad, CA, USA), boiled for 5 min at 95 °C and, eventually, undergone to the electrophoresis. Then, proteins were transferred to a PVDF membrane (Invitrogen, Carlsbad, CA, USA) using iBlot™ 2 Gel Transfer Device (Invitrogen, Carlsbad, CA, USA) and different antibodies were tested. Primary antibodies were all prepared in Flex Solution (iBind™ Flex Solution Kit, Invitrogen, Carlsbad, CA, USA) and were incubated overnight at 4 °C with gentle shaking. The day after, the primary antibody was removed, and the membrane was washed three times with TBS-T. After that, the membrane was incubated for 1 h with the secondary antibody, dissolved in Flex Solution (iBind™ Flex Solution Kit, Invitrogen, Carlsbad, CA, USA). Following three washes in TBST, bands were detected by chemiluminescence (ChemiDoc™ XRS, Biorad, Hercules, CA, USA), rinsing the membrane with ECL reagents (SuperSignal™ West Pico PLUS Chemiluminescent Substrate, Thermo Scientific™, Rockford, IL, USA, and Pierce™ ECL Western Blotting Substrate). Primary and secondary antibodies, and the related dilutions, are reported in the Table 1.

### 4.15. Statistical Analyses 

Each experiment was repeated at least three times. All the statistical analyses were performed with PrismTM computer program (GraphPad, San Diego, CA, USA). Data were analyzed by a one way ANOVA, followed by Dunnett’s test. Statistical significance was determined at α = 0.05 level. Differences were considered statistically significant when *p* < 0.05.

## 5. Conclusions

In conclusion, in this study, we conducted a large series of experiments comparing Rapa and SAP in cellular models of human glioma. Focusing on the diverging effects of the two compounds, perhaps most relevant is the opposite effect on microglia polarization. In this paradigm, SAP clearly drives microglia toward an M2 phenotype, which is commonly thought of as a factor favoring tumor proliferation. The in vitro findings might be possibly related to the lack of efficacy so far shown by SAP in the treatment of human GBM.

## Figures and Tables

**Figure 1 ijms-24-09834-f001:**
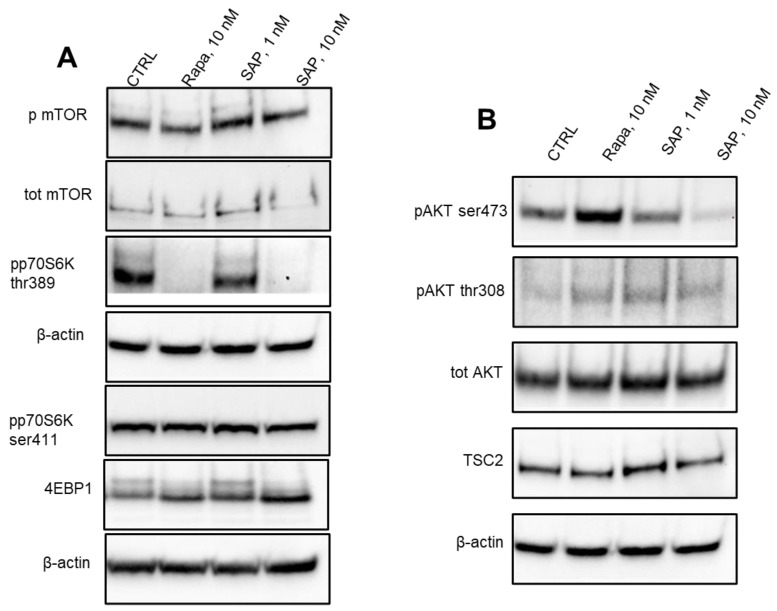
Western blot analysis in U87MG of the main proteins involved in the PI3K/AKT/mTOR pathway after 2 h of the following treatments: Lane 1, control, Lane 2, Rapamycin 10 nM, Lane 3, Sapanisertib 1 nM, Lane 4, Sapanisertib 10 nM. For every protein set, β-actin is reported as the normalizer gene. (**A**): main proteins of mTORC1 complex; (**B**): main proteins of mTORC2 complex.

**Figure 2 ijms-24-09834-f002:**
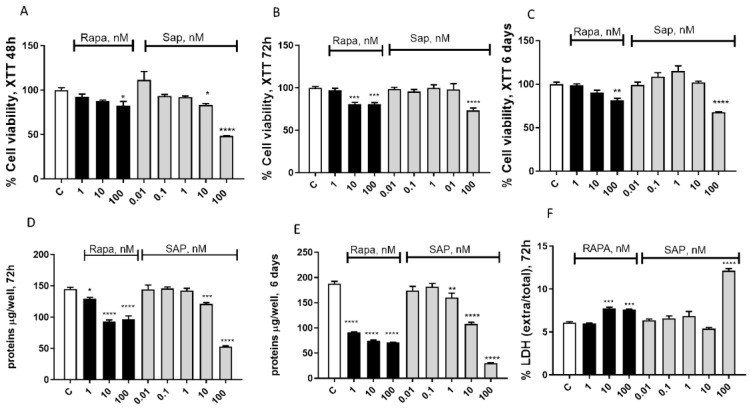
Evaluation of treatment toxicity and the effect on cell survival in U87MG Effect of Rapa, ranged from 1 nM to 100 nM, and SAP, ranged from 0.01 nM to 100 nM on cell viability at 48 h (**A**), 72 h (**B**) and 6 days (**C**) by XTT assay. Data are expressed as a percentage relative to the untreated cells (control = 100%) and are means ± SEM. One-way ANOVA analysis, followed by Dunnett’s post-test, was conducted. * *p* < 0.05, ** *p* < 0.01, *** *p* < 0.005, **** *p* < 0.0001. All *p* values were calculated versus the control sample. (**D**,**E**) show total protein μg per well at 72 h and 6 days, respectively. (**F**) shows extracellular/total LDH ratio in U87 MG treated for 72 h with Rapamycin, ranging from 1 nM to 100 nM, and Sapanisertib, ranging from 0.01 nM to 100 nM. Data are means ± SEM and one-way ANOVA analysis, followed by Dunnett’s post-test, was carried out. * *p* < 0.05, ** *p* <0.01, *** *p* < 0.005, **** *p* < 0.0001. All *p* values were calculated versus the control sample.

**Figure 3 ijms-24-09834-f003:**
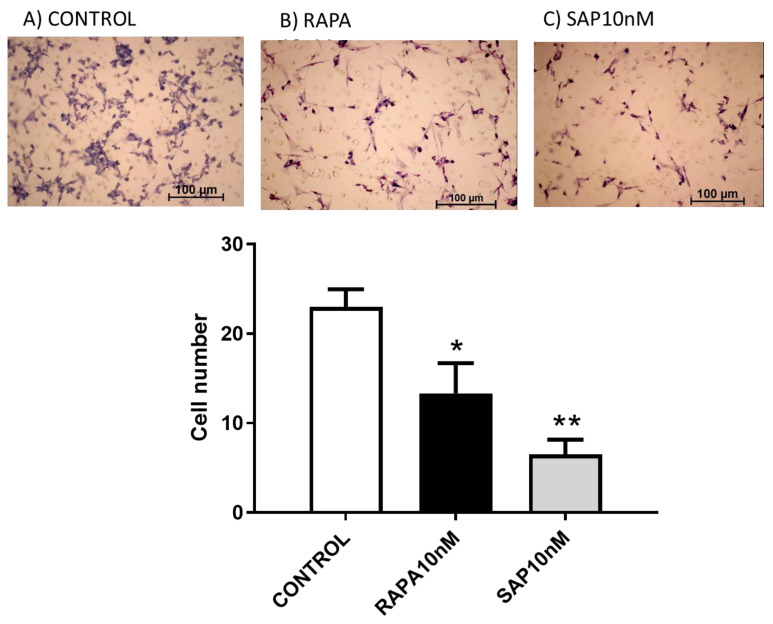
Effect of Rapa and SAP on U87 migration and the potential role in reducing tumor invasiveness. (**A**–**C**) show the untreated cells, cells treated with 10 nM Rapamycin and cells treated with Sapanisertib 10 nM, respectively. Cell number count was calculated as mean ± SEM and one-way ANOVA analysis, followed by Dunnett’s post-test, was carried out. * *p* < 0.05, ** *p* < 0.01.

**Figure 4 ijms-24-09834-f004:**
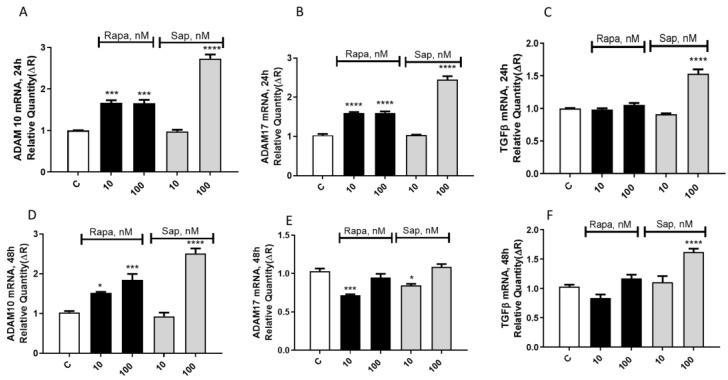
Evaluation of Adam 10 (**A**,**D**), Adam 17 (**B**,**E**) and TGFβ (**C**,**F**) mRNA level expression in U87MG treated with Rapa and SAP, both at 10 and 100 nM. Results refer to 24 and 48 h of treatment. Data are expressed as a fold change of treated samples versus control, considered as a calibrator. Data are means ± SEM, and were analyzed by one-way ANOVA, followed by Dunnett’s post-test. * *p* < 0.05, *** *p* < 0.005, **** *p* < 0.0001.

**Figure 5 ijms-24-09834-f005:**
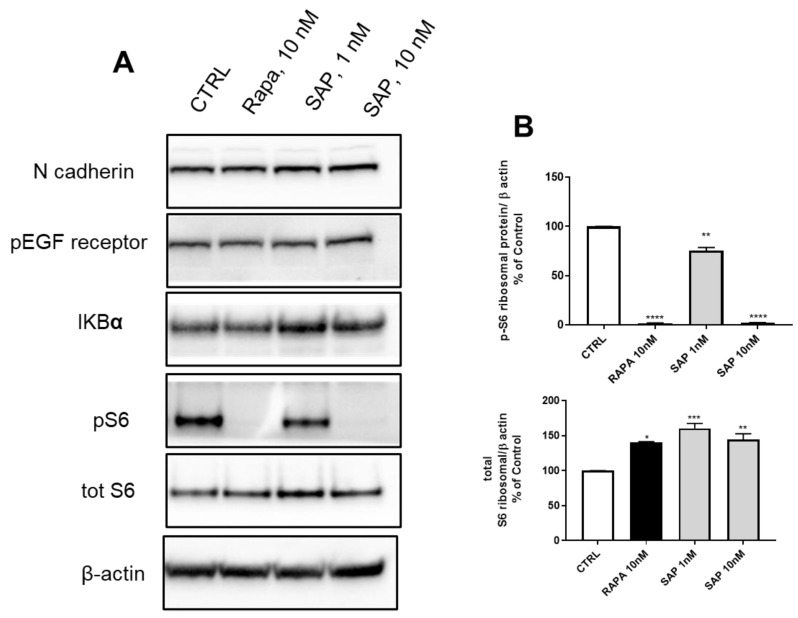
Western blot analysis in U87MG of the main proteins involved in the EMT mechanism after 2 h of the following treatments: Lane 1, control, Lane 2, Rapamycin 10 nM, Lane 3, Sapanisertib 1 nM, Lane 4, Sapanisertib 10 nM. For every protein set, β-actin is reported as the normalizer gene. (**A**) WB bands (**B**): Data are means ± SEM, and were analyzed by one-way ANOVA, followed by Dunnett’s post-test. * *p* < 0.05, ** *p* < 0.005, *** *p* < 0.001, **** *p* < 0.0001.

**Figure 6 ijms-24-09834-f006:**
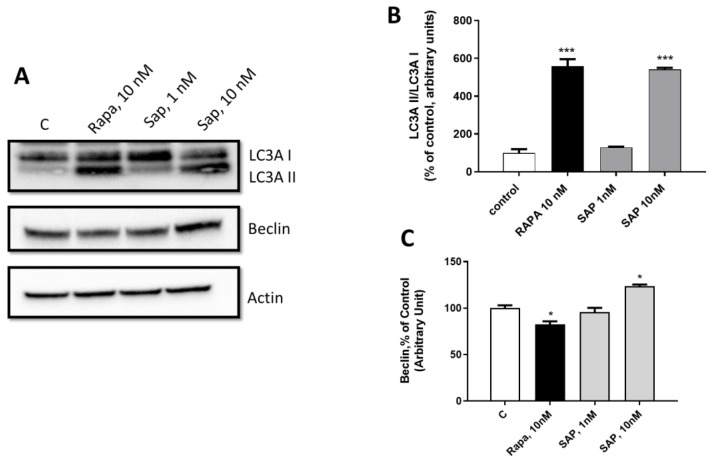
Evaluation of the involvement of SAP in regulating LCA3 and Beclin-1 mediated-autophagy mechanisms in U87MG. (**A**) WB bands (**B**) refers to LC3A II/LC3A ratio. (**C**) shows Beclin-1 level expression when treating with Rapa and SAP. β actin was used as the normalizer gene. Results are expressed as percentage relative to the untreated cells (control = 100%) and are means ± SEM. t Data refer to 6 days of the following treatments: Lane 1, control, Lane 2, Rapamycin 10 nM, Lane 3, Sapanisertib 1 nM, Lane 4, Sapanisertib 10 nM. Data are means ± SEM, and were analyzed by one-way ANOVA, followed by Dunnett’s post-test. * *p* < 0.05, *** *p* < 0.001.

**Figure 7 ijms-24-09834-f007:**
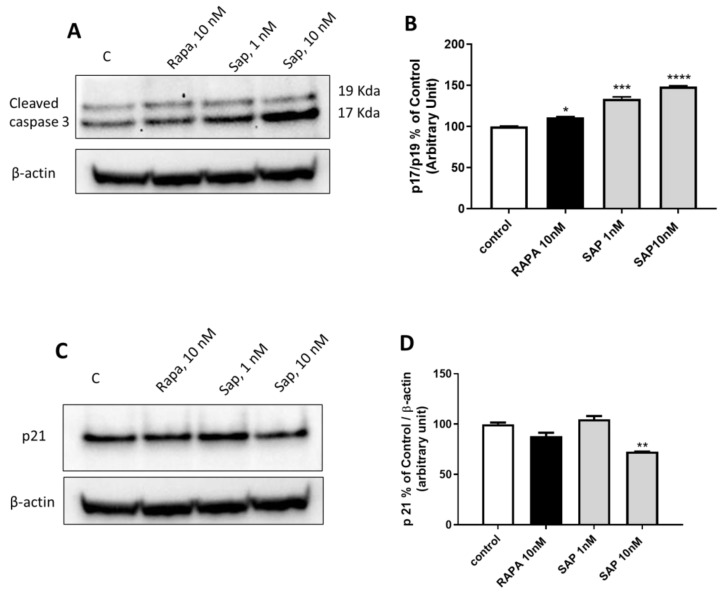
Evaluation of the involvement of SAP in regulating Caspase 3 mediated- apoptosis mechanisms in U87MG. (**A**) refers to P17/P19 ratio. (**B**) shows densitometry of P17 on P19 form. (**C**) refers to p21 protein. (**D**) shows densitometry of p21 on actin. β actin was used as the normalizer gene. Results are expressed as percentage relative to the untreated cells (control = 100%) and are means ± SEM. t Data refer to 2 h of the following treatments: Lane 1, control, Lane 2, Rapamycin 10 nM, Lane 3, Sapanisertib 1 nM, Lane 4, Sapanisertib 10 nM. Data are means ± SEM, and were analyzed by one-way ANOVA, followed by Dunnett’s post-test. * *p* < 0.05, ** *p* < 0.01, *** *p* < 0.001. **** *p* < 0.0001.

**Figure 8 ijms-24-09834-f008:**
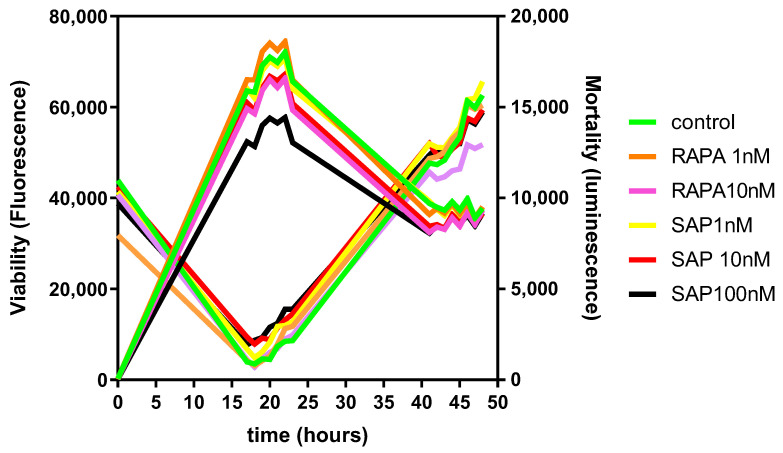
General overview of the effect of Rapamycin and Sapanisertib on cell viability, expressed as a function of fluorescence, (**left axis**) and cell mortality, expressed as a function of luminescence (**right axis**), on CHME-5 in basal conditions. These parameters were measured during 48 h- treatment.

**Figure 9 ijms-24-09834-f009:**
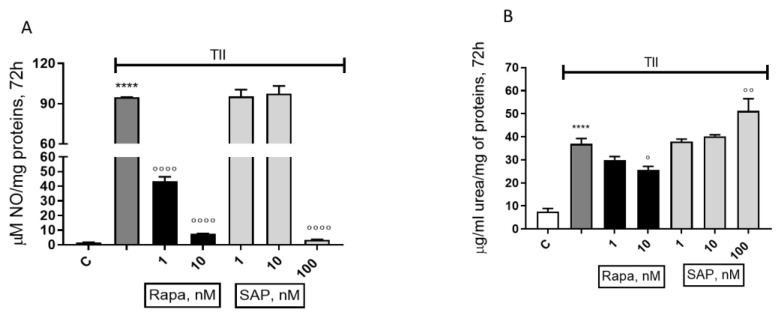
Comparison between the effect of Rapamycin and Sapanisertib on nitrite (**A**) and urea (**B**) release in CHME-5 after a proinflammatory stimulation. The mentioned parameters were measured after 72 h-treatment. Data are means ± SEM, and were analyzed by one-way ANOVA, followed by Dunnett’s post-test. **** *p* < 0.0001 vs control, ° *p* < 0.05, °° *p* < 0.01, °°°° *p* < 0.0001 vs. TII.

**Figure 10 ijms-24-09834-f010:**
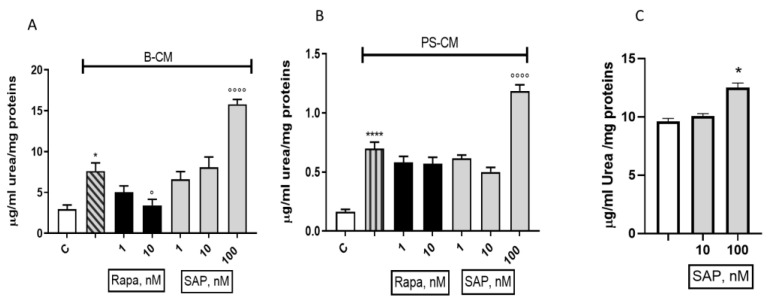
Effect of Rapamycin and Sapanisertib on urea release when CHME-5 acquire M2 phenotype polarization, after stimulation with B-CM (**A**) and PS-CM (**B**). (**C**) shows urea release in co-culture of CHME-5 and U87MG after 48 h of treatment with SAP. Urea amount was measured after 48 h-treatment. Data are means ± SEM, and one-way ANOVA analysis, followed by Bonferroni’s post-test, was conducted. * *p* < 0.05, **** *p* < 0.0001 vs control. ° *p* < 0.05, °°°° *p* < 0.0001 vs. CM.

**Table 1 ijms-24-09834-t001:** Antibodies and related dilutions used in western blotting analysis.

Antibody	Dilution	Producer	Catalog Number
p-mTOR (Ser 2448)	1:1000	Cell Signaling	#2971
total mTOR	1:1000	Cell Signaling	#2972
4EBP1	1:1000	Bethyl	#A300-501A
pp70S6K Thr 389	1:500	Cell Signaling	#9205
pp70S6K Ser 411	1:1000	Santa Cruz	sc-8416
pAKT Ser 473	1:500	Cell Signaling	#4060
pAKT Thr 308	1:500	Cell Signaling	#9275
total AKT	1:1000	ThermoFisher Scientific	#MA5-14916
LC3A	1:250	Novus Bio	#NB100-2331
Beclin-1	1:1000	Novus Bio	#NB500-249
TSC2	1:1000	Cell Signaling	#4308
S6 Ribosomal Protein	1:1000	Cell Signaling	#2217
Phospho-S6 Ribosomal Protein (Ser 235/236)	1:1000	Cell signaling	#2211
Phospho-EGF Receptor (Tyr1068)	1:1000	Cell signaling	#3777
N-Cadherin	1:1000	BD Bioscience	#610920
Cleaved Caspase-3 (Asp175)	1:1000	Cell signaling	#9664
p21 Waf1/Cip1	1:500	Cell Signaling	#2947
IKB-α	1:1000	Santa Cruz	#sc-371
β-actin	1:1000	Sigma	#A 5316
Anti mouse	1:3000	Sigma	#A 3682
Anti rabbit	1:15,000	Jackson Immuno Research	#111-035-045

## Data Availability

Raw data presented in this study are available on request from the corresponding author.

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
