# Peer review of "mTOR Inhibition Is Effective against Growth, Survival and Migration, but Not against Microglia Activation in Preclinical Glioma Models"

_ijms, 2023, doi:10.3390/ijms24129834_

Round 1
Reviewer 1 Report
The authors seek to study the suitability of the mTOR ATP-competitive inhibitor sapanisertib (aka. INK128, MLN0128, or TAK-228) as a drug to potentially treat glioblastoma multiforme. They compare the effect of sapanisertib with the effect of rapamycin in different human glioblastoma cell lines. The authors identify some overlapping effects (e.g., reduction of cell viability or increase in the number of migrating cells) and some non-overlapping (e.g., urea production in microglia cells). Finally, the authors conclude that sapanisertib drives microglia toward a tumor-favoring phenotype that may be related to the lack of efficacy shown by sapanisertib in the treatment of glioblastoma. Specific comments are as follows:
1) Related to Figure 2. The authors assess the effect of the drugs on mTORC1 and mTORC2 signaling after 6 days of treatment (“refreshing” the dose after 3 days of treatment). This seems too long as signaling events occur within minutes and are transient. The authors should assess the effect of the drugs on mTOR signaling after, for example, 30-60 min or several hours of treatment. Also, as the authors comment in the Discussion section, it is surprising that sapanisertib fails to inhibit mTORC1 and mTORC2 – the drug is expected to behave as an ATP competitive inhibitor of mTOR. The authors postulate that “some resistance phenomenon may intervene”. However, the lack of mTORC1 and mTORC2 inhibition by sapanisertib could also be related to when after the treatment the effect is assessed. The authors’ conclusion might be different if they performed the experiment recommended above.
2) Related to Figure 1. What is the rationale behind determining the levels of RAPTOR and RICTOR after drug treatment? Are the authors implying that the differences in RAPTOR and RICTOR levels affect mTOR complex integrity? If yes, they should prove it experimentally by, for example, immunoprecipitating one component (either RAPTOR/RICTOR or mTOR) and determining the levels of the other (either mTOR or RAPTOR/RICTOR). Also, there is a swap between the RAPTOR and RICTOR protein labeling between the panels and the figure legend. The figure legend seems correct (RAPTOR levels are assessed in Panel A and RICTOR levels in Panel B).
3) Figure 2, Panel C. The correct three-letter code for the amino acid Threonine is Thr and not The.
4) Figure 9, line 231. There is a “P” missing in “anel A”
Author Response
The authors wish to thank the referee for careful revision work.
1) Related to Figure 2. The authors assess the effect of the drugs on mTORC1 and mTORC2 signaling after 6 days of treatment (“refreshing” the dose after 3 days of treatment). This seems too long as signaling events occur within minutes and are transient. The authors should assess the effect of the drugs on mTOR signaling after, for example, 30-60 min or several hours of treatment. Also, as the authors comment in the Discussion section, it is surprising that sapanisertib fails to inhibit mTORC1 and mTORC2 – the drug is expected to behave as an ATP competitive inhibitor of mTOR. The authors postulate that “some resistance phenomenon may intervene”. However, the lack of mTORC1 and mTORC2 inhibition by sapanisertib could also be related to when after the treatment the effect is assessed. The authors’ conclusion might be different if they performed the experiment recommended above.
New experiments were conducted. We ran western blotting at 2 hours. The results have been inserted in a new figure 1 and described on page 3, lines 111-122.
We put here the images:
2) Related to Figure 1. What is the rationale behind determining the levels of RAPTOR and RICTOR after drug treatment? Are the authors implying that the differences in RAPTOR and RICTOR levels affect mTOR complex integrity? If yes, they should prove it experimentally by, for example, immunoprecipitating one component (either RAPTOR/RICTOR or mTOR) and determining the levels of the other (either mTOR or RAPTOR/RICTOR). Also, there is a swap between the RAPTOR and RICTOR protein labeling between the panels and the figure legend. The figure legend seems correct (RAPTOR levels are assessed in Panel A and RICTOR levels in Panel B).
Thanks to comments from the referees, we decided to remove figure 1 because, as you rightly pointed out, it does not add important conclusions to the data of the work.
3) Figure 2, Panel C. The correct three-letter code for the amino acid Threonine is Thr and not The.
Thanks, we corrected the typo.
4) Figure 9, line 231. There is a “P” missing in “anel A”
Thanks, we corrected the typo.

Reviewer 2 Report
Effects of dual mTOR inhibitor sapanisertib on human glioma cell lines. Comparison with selective mTORC1 inhibitor rapamycin. by Lisi et al. is an interesting manuscript that lacks novelty. Namely, Sapanisertib is already included in two active Phase II clinical studies for glioblastoma treatment. The authors also mentioned this in the Introduction. They did not emphasize why it is important to study again its molecular mechanisms in vitro. This manuscript's only important finding is that sapanisertib induced an M2 profile in microglia associated with poor clinical outcomes. Therefore, my recommendation is major revision due to:
1. The concept of the manuscript should be changed. Comparison between sapanisertib and rapamycin in two glioblastoma cell lines should not attract the attention of the IJMS readership. Therefore, the more intriguing concept with significant results and a take-home message is that sapanisertib can be associated with poor glioblastoma patients’ outcomes due to the promotion of anti-inflammatory microglia.
2. Introduction lacks the information on sapanisertib’s capability to pass the blood-brain barrier.
3. Discussion is too long with explanations that should be already known to the reader interested in this research field (autophagy, mTOR inhibitors, etc.).
Minor concerns:
1. Figure 1: A and B are mixed (text, legend, Figure)
2. Results with another glioblastoma cell line T98G should be moved to Supplementary (Figures 7, 8, and 9).
3. Table 2. Summary of results is completely unnecessary and therefore very confusing. It should be removed.
4. Table 1. Antibodies… should be Table 2. because it is provided after Table 1. Summary of results.
Author Response
The authors wish to thank the referee for careful revision work.
- The concept of the manuscript should be changed. Comparison between sapanisertib and rapamycin in two glioblastoma cell lines should not attract the attention of the IJMS readership. Therefore, the more intriguing concept with significant results and a take-home message is that sapanisertib can be associated with poor glioblastoma patients’ outcomes due to the promotion of anti-inflammatory microglia.
The title was changed in : “mTOR inhibition is effective against growth, survival and migration, but not against microglia activation in preclinical glioma models.”
- Introduction lacks the information on sapanisertib’s capability to pass the blood-brain barrier.
There are no specific studies that deepen the ability of SAP to cross the blood-brain barrier but now the molecule is studied in different CNS diseases where there is an effect due to the molecule:
Some examples:
Enhancing CDK4/6 inhibitor therapy for medulloblastoma using nanoparticle delivery and scRNA-seq-guided combination with sapanisertib.
Lim C, Dismuke T, Malawsky D, Ramsey JD, Hwang D, Godfrey VL, Kabanov AV, Gershon TR, Sokolsky-Papkov M.Sci Adv. 2022 Jan 28;8(4):eabl5838. doi: 10.1126/sciadv.abl5838. Epub 2022 Jan 26.PMID: 35080986 Free PMC article.
Deficiency in the Treatment Description of mTOR Inhibitor Resistance in Medulloblastoma, a Systematic Review.
Alammar H, Nassani R, Alshehri MM, Aljohani AA, Alrfaei BM.Int J Mol Sci. 2021 Dec 31;23(1):464. doi: 10.3390/ijms23010464.PMID: 35008889 Free PMC article. Review.
Unbiased Proteomic and Phosphoproteomic Analysis Identifies Response Signatures and Novel Susceptibilities After Combined MEK and mTOR Inhibition in BRAFV600E Mutant Glioma.
Maxwell MJ, Arnold A, Sweeney H, Chen L, Lih TM, Schnaubelt M, Eberhart CG, Rubens JA, Zhang H, Clark DJ, Raabe EH.Mol Cell Proteomics. 2021;20:100123. doi: 10.1016/j.mcpro.2021.100123. Epub 2021 Jul 21.PMID: 34298159 Free PMC article.
TORC1/2 kinase inhibition depletes glutathione and synergizes with carboplatin to suppress the growth of MYC-driven medulloblastoma.
Maynard RE, Poore B, Hanaford AR, Pham K, James M, Alt J, Park Y, Slusher BS, Tamayo P, Mesirov J, Archer TC, Pomeroy SL, Eberhart CG, Raabe EH.Cancer Lett. 2021 Apr 28;504:137-145. doi: 10.1016/j.canlet.2021.02.001. Epub 2021 Feb 8.PMID: 33571541 Free PMC article.
mTOR kinase inhibition disrupts neuregulin 1-ERBB3 autocrine signaling and sensitizes NF2-deficient meningioma cellular models to IGF1R inhibition.
Beauchamp RL, Erdin S, Witt L, Jordan JT, Plotkin SR, Gusella JF, Ramesh V.J Biol Chem. 2021 Jan-Jun;296:100157. doi: 10.1074/jbc.RA120.014960. Epub 2020 Dec 9.PMID: 33273014 Free PMC article.
Synergistic activity of mTORC1/2 kinase and MEK inhibitors suppresses pediatric low-grade glioma tumorigenicity and vascularity.
Arnold A, Yuan M, Price A, Harris L, Eberhart CG, Raabe EH.Neuro Oncol. 2020 Apr 15;22(4):563-574. doi: 10.1093/neuonc/noz230.PMID: 31841591 Free PMC article.
- Discussion is too long with explanations that should be already known to the reader interested in this research field (autophagy, mTOR inhibitors, etc.).
In agreement with the referee, in the new version of the manuscript we shortened the discussion and removed table 2.
Minor concerns:
- Figure 1: A and B are mixed (text, legend, Figure).
Thanks for reporting, but in agreement with referees, figure 1 has been removed from the manuscript.
- Results with another glioblastoma cell line T98G should be moved to Supplementary (Figures 7, 8, and 9).
Figure 7, 8 and 9 were moved to supplementary data.
- Table 2. Summary of results is completely unnecessary and therefore very confusing. It should be removed.
In agreement with the referee, in the new version of the manuscript we shortened the discussion and removed table 2.
- Table 1. Antibodies… should be Table 2. because it is provided after Table 1. Summary of results.
Thanks, but now without the table of summary of results the table of antibodies is Table 1.

Reviewer 3 Report
The authors did a lot of work comparing SAP and Rapa in glioma, especially found out that SAP may induce microglia polarized toward a M2 pro-tumor phenotype. The point of SAP on microglia polarization may explain the poor clinial outcome of patients treated with SAP and may provide new evidence of combination of SAP and other medicines may better improving GBM patients survival. However, we think there are some points needed to be revised to better improve the quality of this work.
1. It will be better if the title of this article can combine the two sentences into one summary conclusion.
2. The western figures in the article need to be aligned and adjusted to the same size.
3. The efficacy of various concentrations of Rapa and SAP on Rictor and Raptor in Figure 1 A and B differ from description in the article and the figure legends part. We think maybe the authors make a mistake in labeling the figures or the figures were placed in a wrong order.
4. In the text it is said that phospho-p70 at Ser 411 and at Thr 389 were reduced by 10 nM Rapa in Figure 2A, but the band of actin expression showed similar difference. So it is difficult to tell if these factors downstream the mTOR signaling cascade really changes due to different concentrations of Rapa or just because of sample loading quantity difference.
5. In the text it is said Total Akt was increased by 10nM SAP, but in Figure 2C there is no obvious increasd Total Akt expression.
6. Both of Rapa and SAP are mTOR inhibitors, why total mTOR and factors downstream showed increased expression tendency when cells treated with Rapa and SAP?
7. Figure 3D and Figure 7B only show statistical chart of protein expression. It would be better to show the original western bands.
8. In Figure 5A and B, ADAM 10 and ADAM 17 showed increased expression after cells treated with Rapa and SAP, which was inconsistent with the changes of migratory ability of cells. The authors dicussed this and concluded that ADAM 10 and ADAM 17 not involved in mTOR inhibitor reduction of glioma cell migration. So which key molecules function in mTOR inhibitor reduction of glioma cell migration? Maybe more experiments such as western blot testing more EMT-associated markers functioning in different process of EMT should be done to explore this part.
9. In part 2.1.4, LC3A was miswritten as LCA3
10. In Figure 5 A and B, in U87MG cells ADAM 10 and ADAM 17 showed increased mRNA expression after treated with Rapa and SAP for 24h, but in Figure 9A and B, in T98G cells ADAM 10 and ADAM 17 showed opposite mRNA expression after treated with Rapa and SAP for 6 days. Why not test the expression of ADAM 10 and ADAM 17 in the two cells after the same days of treatment of Rapa and SAP to see if the results remains the same? Also, only mRNA expression were tested here, it would be better to add western bands to directly show protein expression levels.
11. In part 2.3, the effects of mTOR inhibitors in microglia-glioma interaction model lack validation of animal models.
Author Response
The authors wish to thank the referee for careful revision work.
- It will be better if the title of this article can combine the two sentences into one summary conclusion.
The title was changed in: “mTOR inhibition is effective against growth, survival and migration, but not against microglia activation in preclinical glioma models.”
- The western figures in the article need to be aligned and adjusted to the same size.
We aligned and adjusted to the same size the western figures.
- The efficacy of various concentrations of Rapa and SAP on Rictor and Raptor in Figure 1 A and B differ from description in the article and the figure legends part. We think maybe the authors make a mistake in labeling the figures or the figures were placed in a wrong order.
We apologize, yes we made a mistake in naming the panels. However, in the current version of the manuscript the data on raptor and rictor were removed because they were deemed not relevant to the final purpose of the manuscript.
- In the text it is said that phospho-p70 at Ser 411 and at Thr 389 were reduced by 10 nM Rapa in Figure 2A, but the band of actin expression showed similar difference. So it is difficult to tell if these factors downstream the mTOR signaling cascade really changes due to different concentrations of Rapa or just because of sample loading quantity difference.
The data are all reported taking into account normalization on actin, so changes due to different protein concentrations should be zeroed. In this new version of the manuscript as suggested by referee 1 we have inserted wb at a short time, in this new figure the effects on the P70 proteins are more evident.
- In the text it is said Total Akt was increased by 10nM SAP, but in Figure 2C there is no obvious increased Total Akt expression.
In the new experiment (a new time point 2 hours) no difference in AKT content were reported.
- Both of Rapa and SAP are mTOR inhibitors, why total mTOR and factors downstream showed increased expression tendency when cells treated with Rapa and SAP?
In the new figure that included a new time point (2 hours) a reduction after rapa e SAP challenge in p-mTOR and in p-p70 content were reported.
- Figure 3D and Figure 7B only show statistical chart of protein expression. It would be better to show the original western bands.
There are no western blotting bands to show because migration analysis is done with GIEMSA staining and cell count. The original photos are shown in panel A.
- In Figure 5A and B, ADAM 10 and ADAM 17 showed increased expression after cells treated with Rapa and SAP, which was inconsistent with the changes of migratory ability of cells. The authors dicussed this and concluded that ADAM 10 and ADAM 17 not involved in mTOR inhibitor reduction of glioma cell migration. So which key molecules function in mTOR inhibitor reduction of glioma cell migration? Maybe more experiments such as western blot testing more EMT-associated markers functioning in different process of EMT should be done to explore this part.
In the new version of the manuscript we have added a figure in which several proteins involved in EMT have been evaluated. Among these, the S6 protein is significantly reduced. The results have been included on page 7 lines 185-189.
- In part 2.1.4, LC3A was miswritten as LCA3
We apologize for the error, we corrected the spelling.
- In Figure 5 A and B, in U87MG cells ADAM 10 and ADAM 17 showed increased mRNA expression after treated with Rapa and SAP for 24h, but in Figure 9A and B, in T98G cells ADAM 10 and ADAM 17 showed opposite mRNA expression after treated with Rapa and SAP for 6 days. Why not test the expression of ADAM 10 and ADAM 17 in the two cells after the same days of treatment of Rapa and SAP to see if the results remains the same? Also, only mRNA expression were tested here, it would be better to add western bands to directly show protein expression levels.
All timepoints were tested for both cell lines (24-48 hours and 6 days), but we only reported data where the changes were significant. In the new manuscript version, a number of proteins involved in EMT were analyzed in wb.
- In part 2.3, the effects of mTOR inhibitors in microglia-glioma interaction model lack validation of animal models.
In this project we have included a lot of molecular data by investigating different pathways on cell lines. In the continuation of the research, we could think of studying the functional data also in vivo. Thanks for the suggestion.

Reviewer 4 Report
The authors present the manuscript Effects of dual mTOR inhibitor sapanisertib on human glioma cell lines. Comparison with selective mTORC1 inhibitor rapamycin. In this study the authors indicate a non-selective mTOR inhibitor, sapanisertib, has an effect on human glioblastoma cell lines. While the authors perform a dose curve of 1 vs 10nM sapanisertib, much of th antigrowth properties are alleviated after 24-48hours.
The indication is that Rapamycin has no biological effects on proteins such as raptor and mTor, yet can still alter growth rates and induce autophagy. Can the authors address how rapamycin can similarly induce autophagy, yet has not effect on rictor, raptor, or mTor?
Can the author combine Rapamycin and Sapanisertib to reduce cell numbers/ viability.
Have the authors assess additional cellar phenotypes (including cell migration, apoptosis, caspase activity?)
Author Response
The authors wish to thank the referee for careful revision work.
The indication is that Rapamycin has no biological effects on proteins such as raptor and mTor, yet can still alter growth rates and induce autophagy. Can the authors address how rapamycin can similarly induce autophagy, yet has not effect on rictor, raptor, or mTor?
In the original version of manuscript we wrong the time point where we studied mTOR pathway. In the current version we studied mTOR pathway after 2h of treatment and a reduction after Rapa treatment in pospho-mTOR protein and effects also in 4EBP1 was reported (see new figure 1).
Can the author combine Rapamycin and Sapanisertib to reduce cell numbers/ viability.
The goal of our study is to compare the effects of rapamycin, a selective mTOR inhibitor with the effects of a non-selective inhibitor such as sapanisertib. The goal of another study could be to see if the two molecules have additive or potentiating powers.
Have the authors assess additional cellar phenotypes (including cell migration, apoptosis, caspase activity?)
Yes, cell migration data are included (Figures #3-4-5).

Round 2
Reviewer 1 Report
The authors have successfully addressed all issues raised. Specific comments are as follows:
1) Related to Figure 1A. The authors conclude that “4EBP1 was increased by 10 nM Rapa and 10 nM SAP”. I am not sure how relevant this conclusion is. The authors should instead focus on the phosphorylation status of 4EBP1. When using an antibody against 4EBP1 (as is the case of the authors) phosphorylation can be assessed by analyzing the slower migrating bands. The authors should mention that 4EBP phosphorylation is reduced by 10 nM Rapa and 10 nM SAP and that this is in agreement with what they observe for S6K.
2) Related to Figure 1B. It is not surprising that mTORC2 activity increases (in other words, phosphorylation of AKT at Ser 473 increases) after 2 h of rapamycin treatment. This is very likely due to the release of the negative feedback loop formed by the phosphorylation and inhibition of IRS1, and thereby PI3K, by S6K (reviewed, e.g., in PMID: 24556838). The authors should mention and discuss this.
3) Related to Figure 1A, table 1, and text (lines 115-117). When referring to the ribosomal protein S6 kinase, I suggest the authors should name it S6K instead of p70. This would avoid confusion as S6K is used more commonly than p70.
4) Lines 119-122. The authors should revise the wording. “TOT AKT” should be replaced by total AKT. Also, the sentence “the levels … was not modify” seems ungrammatical.
5) Lines 211-212. The authors state that the main experiments were performed in U87MG cells “and repeated in another human glioma cell line, T98G” and refer the reader to the Supplementary Material for an explanation. It would be more informative if in the main text they wrote “obtaining similar results” after T98G.
6) Table 1. Please include the catalog number for each antibody and, if applicable, the phosphorylation site. Also, for the Phospho-S6 and Phospho-EGF receptor antibodies, in the “Producer” column it is written “Cell”. Is this correct or should it be “Cell Signaling”?
Author Response
The authors wish to thank the referee for careful revision work.
1) Related to Figure 1A. The authors conclude that “4EBP1 was increased by 10 nM Rapa and 10 nM SAP”. I am not sure how relevant this conclusion is. The authors should instead focus on the phosphorylation status of 4EBP1. When using an antibody against 4EBP1 (as is the case of the authors) phosphorylation can be assessed by analyzing the slower migrating bands. The authors should mention that 4EBP phosphorylation is reduced by 10 nM Rapa and 10 nM SAP and that this is in agreement with what they observe for S6K.
As suggested by the referee we have modified the description of the results, giving greater emphasis to the change in phosphorylation of 4EBP1. Page 3 line 111-112.
2) Related to Figure 1B. It is not surprising that mTORC2 activity increases (in other words, phosphorylation of AKT at Ser 473 increases) after 2 h of rapamycin treatment. This is very likely due to the release of the negative feedback loop formed by the phosphorylation and inhibition of IRS1, and thereby PI3K, by S6K (reviewed, e.g., in PMID: 24556838). The authors should mention and discuss this.
The consideration of referee was added and the paper cited page 3 line 120-125.
3) Related to Figure 1A, table 1, and text (lines 115-117). When referring to the ribosomal protein S6 kinase, I suggest the authors should name it S6K instead of p70. This would avoid confusion as S6K is used more commonly than p70.
In the text, p70 has been replaced by pp70S6K.
4) Lines 119-122. The authors should revise the wording. “TOT AKT” should be replaced by total AKT. Also, the sentence “the levels … was not modify” seems ungrammatical.
Grammatical and typing errors have been corrected.
5) Lines 211-212. The authors state that the main experiments were performed in U87MG cells “and repeated in another human glioma cell line, T98G” and refer the reader to the Supplementary Material for an explanation. It would be more informative if in the main text they wrote “obtaining similar results” after T98G.
We added “obtaining similar results”.
6) Table 1. Please include the catalog number for each antibody and, if applicable, the phosphorylation site. Also, for the Phospho-S6 and Phospho-EGF receptor antibodies, in the “Producer” column it is written “Cell”. Is this correct or should it be “Cell Signaling”?
Thanks for the suggestion, catalog numbers and phosphorylation sites were added.
Reviewer 2 Report
The authors made an effort and significantly improved their manuscript. There are only two minor suggestions:
1. Sapanisertib’s ability to pass the blood-brain barrier can be elucidated using the SwissADME online tool. The authors can add this evidence in the Supplementary. It is strange that this drug has been used in clinical trials for glioblastoma patients without this essential information in the literature.
2. Please remove the subtitle: 2.2. Effects of mTOR inhibitors on T98G cells.
Comment on the results obtained on T98G cells together with the effects on U87 cells and refer to Supplementary Figures.
Author Response
The authors wish to thank the referee for careful revision work.
- Sapanisertib’s ability to pass the blood-brain barrier can be elucidated using the SwissADME online tool. The authors can add this evidence in the Supplementary. It is strange that this drug has been used in clinical trials for glioblastoma patients without this essential information in the literature.
We are not experts in the software suggested by the referee (SwissADME). However, we tried to use it and below are the results.
In parallel from a careful study of the literature, we found a study conducted in mice where Sap levels were measured centrally. Below are the results reported in the study of Parkhurst, A., Wang, S.Z., Findlay, T.R. et al. Dual mTORC1/2 inhibition compromises cell defenses against exogenous stress potentiating Obatoclax-induced cytotoxicity in atypical teratoid/rhabdoid tumors. Cell Death Dis 13, 410 (2022). https://doi.org/10.1038/s41419-022-04868-9.
TAK-228 is rapidly absorbed and crosses the blood–brain barrier in concentrations exceeding the therapeutic threshold required to inhibit the mTOR pathway in vitro (Fig. 7A). Concentrations of TAK-228 in plasma and peripheral organs were 4x and 100x higher than in the CNS, respectively (Supplementary Fig. 5)
The above paper was cited in the introduction of our work, page 2 lines 72-78.
- Please remove the subtitle: 2.2. Effects of mTOR inhibitors on T98G cells.
The subtitle is removed.
Comment on the results obtained on T98G cells together with the effects on U87 cells and refer to Supplementary Figures.
A period in discussion section was added page 10 lines 337-340.

Reviewer 3 Report
1. In line 112, phospho-mTOR antibody from CST should be ser 2448, not ser 244.
2. In new Figure 1A, the band of actin looks equal.
3. In new Figure 1A, effects of inhibitors on pp70 thr389 are more evident, not pp70 ser411, please explain the reason.
4. In new Figure 5A&5B, pS6 ribosomal protein was significantly reduced after 2h treatment of Rapa or SAP. please explain the reason why other EMT-related protein does not change.
5. The effects of mTOR inhibitors in microglia-glioma interaction model lack validation of animal models.
Author Response
The authors wish to thank the referee for careful revision work.
- In line 112, phospho-mTOR antibody from CST should be ser 2448, not ser 244.
Typo has been corrected.
- In new Figure 1A, the band of actin looks equal.
In the old figure 2 (now figure 1) we have two bands of actin that come from the same experiment but from two different electrophoretic runs for wb. For this reason, we preferred to report both even if they are rightly very similar to each other.
- In new Figure 1A, effects of inhibitors on pp70 thr389 are more evident, not pp70 ser411, please explain the reason.
This finding is not surprising in fact the phosphorylation on threonine is more and directly controlled by the activity of mTOR.
- Saxton, R.A.; Sabatini, D.M. mTOR Signaling in Growth, Metabolism, and Disease. 2017, 168:960 976.
- Zhou, H.; Huang, S. Role of mTOR signaling in tumor cell motility, invasion and metastasis. Curr Protein Pept Sci. 2011,12:30-42.
- Hou, Z.; He, L.; Qi, R.Z. Regulation of s6 kinase 1 activation by phosphorylation at ser-411. J Biol Chem. 2007, 282:69228
- In new Figure 5A&5B, pS6 ribosomal protein was significantly reduced after 2h treatment of Rapa or SAP. please explain the reason why other EMT-related protein does not change.
Biochemically and rationally, not always all the proteins involved in an event move at the same time. Other proteins may have modification at different times or be involved to different degrees. The drugs we are administering are drugs that impact with mTOR pathway, so it is plausible that the modification of the S6 protein is one of the first movements that correlate with the reduction of migration.
- The effects of mTOR inhibitors in microglia-glioma interaction model lack validation of animal models.
We fully agree with the referee: there is a lack of the animal model but as previously answered our work is conducted essentially at the cellular and molecular level, and having these data was still time- and cost- consuming. The possibility of putting on an animal model could be the conception and departure for a new research project.
Reviewer 4 Report
The authors present the manuscript TmTOR inhibition is effective against growth, survival and migration, but not against microglia activation in preclinical glioma models
The authors wish to thank the referee for careful revision work.
The indication is that Rapamycin has no biological effects on proteins such as raptor and mTor, yet can still alter growth rates and induce autophagy. Can the authors address how rapamycin can similarly induce autophagy, yet has not effect on rictor, raptor, or mTor?
In the original version of manuscript we wrong the time point where we studied mTOR pathway. In the current version we studied mTOR pathway after 2h of treatment and a reduction after Rapa treatment in pospho-mTOR protein and effects also in 4EBP1 was reported (see new figure 1)
THANK YOU FOR MAKING CHANGES TO THE MANUSCRIPT
In figure 1, there is no Rapa at 1nM
In figure 1, there is no quantification of Rictor or Raptor.
In figure 2, there is no quantification of pAKT levels/ total AKT, pmTOR/total mTOR , etc
Can the author combine Rapamycin and Sapanisertib to reduce cell numbers/ viability.
The goal of our study is to compare the effects of rapamycin, a selective mTOR inhibitor with the effects of a non-selective inhibitor such as sapanisertib. The goal of another study could be to see if the two molecules have additive or potentiating powers.
PLEASE MAKE THIS CHANGE TO THE MANUSCRIPT
It is clear Rapa ad Sapa have much of the same effects.
As simple experiment of 1nM + 1nM, 5nm+5nM, and 5nM +10nM , 10nM + 5nM rapa and sapa doses, respectively could be used .
Have the authors assess additional cellar phenotypes (including cell migration, apoptosis, caspase activity?)
Yes, cell migration data are included (Figures #3-4-5).
PLEASE MAKE THIS CHANGE TO THE MANUSCRIPT
There is no adequate measure of apoptosis.
What is the mechanism by which LC3A II modulation mediates autophagy as opposed to changes in LC3AI
Author Response
The authors wish to thank the referee for careful revision work.
In figure 1, there is no Rapa at 1 nM .
All functional data show that effective doses of rapamycin are greater than 1 nM, so having to choose a dose we preferred (for all wb analysis) to choose the 10 nM dose.
In figure 1, there is no quantification of Rictor or Raptor.
As a result of the first round of review, the data on raptor and rictor proteins were deleted because they were assessed as unimportant at the end of the paper itself.
In figure 2, there is no quantification of pAKT levels/ total AKT, pmTOR/total mTOR , etc.
A new supplementary figure showing the normalization graphs pmTOR/mTOR and pAKT/ AKT has been added (new suppl figure 1).
It is clear Rapa ad Sapa have much of the same effects.
As simple experiment of 1nM + 1nM, 5nm+5nM, and 5nM +10nM , 10nM + 5nM rapa and sapa doses, respectively could be used .
Thanks to the suggestion of the referee we conducted combination experiments (Rapa + Sap) and measured proteins and the formation of formazan salt. Data were obtained after 6 days of treatment. A new figure (new suppl figure 2) has been added.
There is no adequate measure of apoptosis.
What is the mechanism by which LC3A II modulation mediates autophagy as opposed to changes in LC3AI.
We know that not all aspects of triggering and maintaining apoptosis have been studied, but we wanted to highlight only the proteins most involved with mTOR.
Round 3
Reviewer 4 Report
In figure 1, there is no Rapa at 1 nM .
All functional data show that effective doses of rapamycin are greater than 1 nM, so having to choose a dose we preferred (for all wb analysis) to choose the 10 nM dose.
OKAY
In figure 1, there is no quantification of Rictor or Raptor.
As a result of the first round of review, the data on raptor and rictor proteins were deleted because they were assessed as unimportant at the end of the paper itself.
OKAY
In figure 2, there is no quantification of pAKT levels/ total AKT, pmTOR/total mTOR , etc.
A new supplementary figure showing the normalization graphs pmTOR/mTOR and pAKT/ AKT has been added (new suppl figure 1).
OKAY
It is clear Rapa ad Sapa have much of the same effects.
As simple experiment of 1nM + 1nM, 5nm+5nM, and 5nM +10nM , 10nM + 5nM rapa and sapa doses, respectively could be used .
Thanks to the suggestion of the referee we conducted combination experiments (Rapa + Sap) and measured proteins and the formation of formazan salt. Data were obtained after 6 days of treatment. A new figure (new suppl figure 2) has been added.
OKAY
There is no adequate measure of apoptosis.
NOT OKAY. Please add additional evidence that supports or refutes a role for apoptosis in your study design. Autophagy is not the same as apoptosis. Have the authors assessed caspase levels or Annexin V
What is the mechanism by which LC3A II modulation mediates autophagy as opposed to changes in LC3AI.
We know that not all aspects of triggering and maintaining apoptosis have been studied, but we wanted to highlight only the proteins most involved with mTOR.
Author Response
The authors wish to thank the referee for careful revision work.
There is no adequate measure of apoptosis. Please add additional evidence that supports or refutes a role for apoptosis in your study design. Autophagy is not the same as apoptosis. Have the authors assessed caspase levels or Annexin V.
As suggested by the referee we ran a WB to evaluate cleaved Caspase 3, in particular we evaluated the ratio between P17/P19 forms. The results have been included on page 8 Lines 217-220 as new figure 7A and B. For consideration of the referee, we report below the figure.

Round 4
Reviewer 4 Report
Response to author query
As suggested by the referee we ran a WB to evaluate cleaved Caspase 3, in particular we evaluated the ratio between P17/P19 forms. The results have been included on page 8 Lines 217-220 as new figure 7A and B. For consideration of the referee, we report below the figure.
Thank you for your response. Because human procaspase-3 is a 32 kDa protein. Please include a blot that shows cleaved caspase-3 ratios as compared to total caspase. That is a more standard method of showing apoptosis data.
Author Response
The authors wish to thank the referee for careful revision work.
Thank you for your response. Because human procaspase-3 is a 32 kDa protein. Please include a blot that shows cleaved caspase-3 ratios as compared to total caspase. That is a more standard method of showing apoptosis data.
Unfortunately, we do not have the required antibody in the lab and we are unable to make purchases. However, in lab we have p21 which as is known is directly connected to procaspase 3 and in particular an increase of p21 means a block of apoptosis. We therefore ran a wb at 2 hours and evaluated the p21 protein. The results were inserted on page 8 Lines 220-227 and two new panels of Figure 7 were added.
Below for considerations of the referee the figure.
